

# Harmonization and comparison of vertically resolved atmospheric state observations: Methods, effects, and uncertainty budget

Arno Keppens[1], Steven Compernolle[1], Tijl Verhoelst[1], Daan Hubert[1], and Jean-Christopher Lambert[1]

[1]Royal Belgian Institute for Space Aeronomy (BIRA-IASB), 1180 Brussels, Belgium

**Correspondence:** Arno Keppens (arno.keppens@aeronomie.be)

**Abstract.** Many applications of atmospheric composition and climate data involve the comparison or combination of vertically resolved atmospheric state variables. Calculating differences and combining data require harmonization of data representations in terms of physical quantities and vertical sampling at least. If one or both datasets result from a retrieval process, knowledge of prior information and averaging kernel matrices in principle allows accounting for retrieval differences as well. Spatiotemporal
mismatch of the sensed air masses and its contribution to the data discrepancies can be estimated with chemistry-transport modelling support. In this work an overview of harmonization or 'matching' operations for atmospheric profile observations is provided. The effect of these manipulations on the information content of the original data and on the uncertainty budget of data comparisons is examined and discussed.

# 1 Introduction

The quality assessment and validation of atmospheric state observations largely relies on making comparisons with (reference) measurements of the same observable. On the other hand, data merging or fusion schemes involve the combination of observations from different sources, weighted by functions that mix uncertainties, information content aspects, and spatiotemporal (4D) representativeness. And chemical data assimilation involves the comparison and/or combination of observations with
modelling outputs. However, quantitative comparisons and combinations of atmospheric soundings are only possible when the observables are represented on the same vertical grid, within the same vertical range, and in identical units. Moreover, observations by different instruments also differ in their sensitivity to and representativeness of spatiotemporal features of the atmospheric field (i.e. resolution or smoothing differences) (Loew et al., 2017). The remote sensing of the atmosphere by space-borne and ground-based instruments additionally consists of under-constrained inverse problems that mix necessary
prior information into the retrieved atmospheric state profiles (Rodgers, 2000). Taking into account these differences in representation, location, and possibly retrieval characteristics is needed for proper data combinations and comparative validation exercises using difference statistics and $\chi^2$ testing (Rodgers and Connor, 2003; von Clarmann, 2006).



Carried out in the context of several satellite validation studies (Sentinel-5p TROPOMI, ESA CCI, EUMETSAT AC-SAF) and of the exploration of advanced data fusion methods (EC H2020 AURORA), with a view to harmonize practices across satellite missions and Earth Observation domains, this work is meant to provide an overview of existing approaches that allow estimating and potentially (partially) correcting for these observational differences in quantitative data comparisons.

The uncertainties that are tied to these differences, as typically expressed in terms of covariance matrices, as a result are also (partially) removed from the uncertainty budget of the data comparison. All relevant difference error contributions are identified in the next section, where it has also been necessary to align some concepts and terminology that might not be identical across all atmospheric research communities. Section 3 then motivates why the difference error contributions must ideally be either quantified or corrected for in the difference statistics. Opting for the latter, an overview of harmonization (or homogenization)

operations that match two atmospheric state datasets in terms of their representation, retrieval characteristics, and location is provided in Section 4. This section focuses on the harmonization algebra, while the reader is referred to the bibliography for specific examples using real data. The impact of the 'matching' operations on the observations' information content and on the comparison uncertainty budget is discussed in Sections 5 and 6 respectively.

## 2   Difference error identification

When taking the difference of two vertically resolved atmospheric state observations, e.g. a measurement under study $\boldsymbol{x}_s$ and a reference measurement $\boldsymbol{x}_r$ that both aim for the same true state $\boldsymbol{x}_t$ as the measurand, random and systematic measurement errors $\boldsymbol{\epsilon}$ on both observations will lead to a non-zero difference vector $\boldsymbol{\Delta\epsilon}$:

$$\boldsymbol{\Delta x} = \boldsymbol{x}_s - \boldsymbol{x}_r = (\boldsymbol{x}_t + \boldsymbol{\epsilon}_s) - (\boldsymbol{x}_t + \boldsymbol{\epsilon}_r) = \boldsymbol{\epsilon}_s - \boldsymbol{\epsilon}_r = \boldsymbol{\Delta\epsilon} \tag{1}$$

This equation only holds for observations that are exactly spatiotemporally co-located. Usually however the air masses that

are sampled by both measurements do not match. This introduces a spatiotemporal (4D) co-location mismatch error, which can be subdivided into a sampling difference error $\boldsymbol{\epsilon}_{\Delta sa}$ (different nominal measurement location and time) and a smoothing difference error $\boldsymbol{\epsilon}_{\Delta sm}$ (different 4D air mass sensitivity) (Nappo et al., 1982; Lambert et al., 2013; Verhoelst et al., 2015). Assuming that both types of error are independent, the above becomes:

$$\boldsymbol{\Delta x} = \boldsymbol{\Delta\epsilon} + \boldsymbol{\epsilon}_{\Delta sa} + \boldsymbol{\epsilon}_{\Delta sm} \tag{2}$$

For vertically resolved atmospheric state observations, a distinction can be made between the horizontal and vertical sampling and smoothing difference errors, next to their temporal counterparts, $\boldsymbol{\epsilon}_{\Delta sa} = \boldsymbol{\epsilon}_{\Delta Hsa} + \boldsymbol{\epsilon}_{\Delta Vsa} + \boldsymbol{\epsilon}_{\Delta Tsa}$ and $\boldsymbol{\epsilon}_{\Delta sm} = \boldsymbol{\epsilon}_{\Delta Hsm} + \boldsymbol{\epsilon}_{\Delta Vsm} + \boldsymbol{\epsilon}_{\Delta Tsm}$, so that:

$$\boldsymbol{\Delta x} = \boldsymbol{\Delta\epsilon} + \boldsymbol{\epsilon}_{\Delta Hsa} + \boldsymbol{\epsilon}_{\Delta Vsa} + \boldsymbol{\epsilon}_{\Delta Tsa}$$
$$+ \boldsymbol{\epsilon}_{\Delta Hsm} + \boldsymbol{\epsilon}_{\Delta Vsm} + \boldsymbol{\epsilon}_{\Delta Tsm} \tag{3}$$

If at least one of the observations is the result of a retrieval process, some retrieval contributions to the difference errors can

be made explicit as well. For example each retrieved profile $\boldsymbol{x}$ that is obtained by an optimal estimation (OE) approach can be



regarded as a weighted average between prior and measurement information (Rodgers, 2000):

$$\boldsymbol{x} = \boldsymbol{A}\boldsymbol{x}_t + (\boldsymbol{I} - \boldsymbol{A})\boldsymbol{x}_p + \boldsymbol{\epsilon} \tag{4}$$

where $\boldsymbol{\epsilon}$ includes, next to (spectral) measurement errors, remote sounding errors like the retrieval parameter errors and forward model errors (Rodgers, 2000, Eq. (3.16)). The latter as such also capture the uncertainty on the square weighting matrix $\boldsymbol{A}$. This
is the so-called averaging kernel matrix (AKM) that is determined by the prior profile shape (PS) $\boldsymbol{x}_p$, the prior constraint (PC) in terms of its covariance matrix $\boldsymbol{S}_p$, and the retrieval process that yields a vertical smoothing and a measurement weight (MW) (also see next sections). Matrix $\boldsymbol{I}$ represents the identity matrix equal in size to the AKM. The following sections however are also valid for retrieval approaches that have $\boldsymbol{x}_p = \boldsymbol{0}$, like in some Philips-Tikhonov regularization schemes, as the equations can easily be adopted accordingly. By inclusion of $\boldsymbol{\epsilon}_{\Delta PS} + \boldsymbol{\epsilon}_{\Delta PC} + \boldsymbol{\epsilon}_{\Delta MW}$ as retrieval difference errors, the observed difference
$\boldsymbol{\Delta x}$ containing at least one retrieved product can be decomposed as follows:

$$\begin{aligned} \boldsymbol{\Delta x} = &\ \boldsymbol{\Delta\epsilon}' + \boldsymbol{\epsilon}_{\Delta Hsa} + \boldsymbol{\epsilon}_{\Delta Vsa} + \boldsymbol{\epsilon}_{\Delta Tsa} \\ &+ \boldsymbol{\epsilon}_{\Delta Hsm} + \boldsymbol{\epsilon}_{\Delta Vsm} + \boldsymbol{\epsilon}_{\Delta Tsm} \\ &+ \boldsymbol{\epsilon}_{\Delta PS} + \boldsymbol{\epsilon}_{\Delta PC} + \boldsymbol{\epsilon}_{\Delta MW} \end{aligned} \tag{5}$$

Here $\boldsymbol{\Delta\epsilon}'$ then contains both the known and unknown measurement errors and remote sounding errors (Rodgers, 2000; Povey and Grainger, 2015).

## 3   Quantitative validation

The Committee on Earth Observation Satellites (CEOS) defines validation as (1) "the process of assessing, by independent means, the quality of the data products" (International Organization for Standardization, 2014). Validation is also defined by international normalization bodies as (2) "the confirmation, through the provision of objective evidence, that specified requirements, adequate for an intended use, have been fulfilled" (Joint Committee for Guides in Metrology, 2012). In the atmospheric remote sensing literature, the validation exercise is also sometimes defined as (3) "to confirm that the theoretical
characterization and error analysis actually represent the properties of the real data" (Rodgers, 2000) or (4) "to confirm the predicted accuracy estimator of that product" (von Clarmann, 2006). The predicted (or inductive or ex-ante) uncertainty of the product under study is typically represented by an error covariance matrix $\boldsymbol{S}_s = \langle \boldsymbol{\epsilon}_s \boldsymbol{\epsilon}_s^T \rangle$, which means that uncertainty information is restricted to covariances and higher order correlations are ignored. Two approaches are commonly applied in order to validate this uncertainty by comparison of the product under study with a reference data product that is characterized
by an ex-ante uncertainty $\boldsymbol{S}_r$.

One can perform a so-called $\chi^2$ test to verify whether the difference between the study and reference products $\boldsymbol{\Delta x}$ is consistent ($\chi^2 \sim 1$) with the predicted estimate of the total uncertainty on the difference $\boldsymbol{S}_\Delta$ (Rodgers and Connor, 2003; von Clarmann, 2006):

$$\chi^2 = L^{-1} \boldsymbol{\Delta x}^T \boldsymbol{S}_\Delta^{-1} \boldsymbol{\Delta x} \tag{6}$$





whereby $L$ symbolizes the number of elements in $\boldsymbol{\Delta x}$ and $\boldsymbol{S_\Delta}$ is the full sum of the covariance matrices of the errors that were previously identified, including $\boldsymbol{S}(\boldsymbol{\Delta\epsilon}) = \boldsymbol{S}_s + \boldsymbol{S}_r$ as the ex-ante uncertainty prediction of the study and reference products combined:

$$\boldsymbol{S_\Delta} = \boldsymbol{S}_s + \boldsymbol{S}_r + \boldsymbol{S}_{\Delta Hsa} + \boldsymbol{S}_{\Delta Vsa} + \boldsymbol{S}_{\Delta Tsa}$$
$$+ \boldsymbol{S}_{\Delta Hsm} + \boldsymbol{S}_{\Delta Vsm} + \boldsymbol{S}_{\Delta Tsm}$$
$$+ \boldsymbol{S}_{\Delta PS} + \boldsymbol{S}_{\Delta PC} + \boldsymbol{S}_{\Delta MW} \tag{7}$$

This expression assumes that the covariances of the difference error terms are not considered in the ex-ante covariance matrices, and should be corrected for those who are. For an ensemble of $N$ pairs of spatiotemporally co-located study and reference profiles, one can now either determine $\chi_N^2 = N^{-1}\sum_n \chi_n^2$ or one can replace the factors in Eq. (6) by statistical estimators. In the latter case a distinction between bias $\boldsymbol{b}$ (systematic) and precision $\boldsymbol{p}$ (random) tests can be made, whereby the combined root-mean-square uncertainty $\boldsymbol{a}$ as an estimator of the accuracy obeys $\boldsymbol{a}^2 = \boldsymbol{b}^2 + \boldsymbol{p}^2$ (von Clarmann, 2006; Joint Committee for
Guides in Metrology, 2008, 2012).

   Secondly, one often directly quantitatively or qualitatively verifies whether a sample bias $\langle\boldsymbol{\Delta x}\rangle$ as an estimator of the combined systematic error of the products is of the same order as the combined ex-ante uncertainty on the mean difference, and whether the corresponding sample dispersion on the differences $\sigma(\langle\boldsymbol{\Delta x}\rangle)$ as an estimator of the standard deviation around the bias (combined random uncertainty) is of the same order as the combined ex-ante random uncertainty on the difference.
Unfortunately it is often overlooked that also here the combined random and systematic components of all difference error contributions to $\boldsymbol{S_\Delta}$ should actually be taken into account, and not only those inductively provided with the study and reference data products through $\boldsymbol{S}_s$ and $\boldsymbol{S}_r$ respectively.

   Irrespective of the method used, a full assessment and quantification of all difference error contributions to $\boldsymbol{S_\Delta}$ is necessary to close the uncertainty budget and perform a proper comparative validation. Alternatively however, one can reduce the (num-
ber of) difference error terms by applying harmonization operations on the study and/or reference profiles. Using matching manipulations, a difference $\boldsymbol{\Delta x}$ is thereby replaced by a difference $\boldsymbol{\Delta x}'$ that contains less or at least reduced difference error contributions. These omitted contributions have then no longer to be considered in $\boldsymbol{S_\Delta'}$ either. On the other hand, note that profile matching operations on their turn introduce manipulation (difference) errors and possibly ancillary data uncertainties.

## 4   Harmonization methods

This section provides an overview of profile matching manipulations. A distinction is made between representation matching (relating to vertical sampling, vertical quantities and their units), vertical sampling matching (cf. grid levels or layers), vertical smoothing matching (cf. vertical resolution of the measurement), retrieval matching (cf. impact of prior information), and spatiotemporal co-location matching. Because of the focus on vertically resolved atmospheric state observations, horizontal and vertical sampling and smoothing issues are discussed separately.



### 4.1 Vertical representation matching (mandatory)

Although not yet related to the error terms in Eq. (5), the matching of the vertical representation of the study and reference profiles is an unavoidable operation to make difference calculations possible in the first place. The vertical representation includes the vertical coordinate (altitude, pressure, geopotential height, or other) and quantity (volume mixing ratio, number

density, partial column, or other). While the former has no impact on the difference (error), except perhaps through its minimal effect on the location of levels or layers, the latter may introduce both manipulation (difference) and ancillary data uncertainties, which actually should be taken into account in the comparison uncertainty budget (see Section 6). In practice however these additional error contributions are often considered to be negligible.

#### 4.1.1 Vertical quantity matching

When changing between concentration-type quantities like number density and volume mixing ratio, a diagonal level-by-level unit conversion matrix $M$ can be constructed straightforwardly (e.g., Keppens et al., 2015, Table B1). The quantity matching operation for a vertical profile $x$ with corresponding ex-ante covariance matrix $S$ and possibly averaging kernel matrix $A$ is then easily achieved by matrix multiplication (Keppens et al., 2015):

$$x' = Mx, S' = MSM^T, A' = MAM^{-1} \tag{8}$$

Note that these operations have no effect on the fractional covariance matrix, nor on the fractional (or logarithmic) averaging kernel matrix that is required for information content studies (see Keppens et al. (2015) and Section 5).

When going from a concentration-type representation on levels to one between levels (i.e. on layers, like partial columns), one can choose the integration boundaries either on the given levels or in between them except for the outer edges, resulting in a rectangular or square conversion matrix $M$, respectively (Keppens et al., 2015, Table B1):

$$M_{(L-1) \times L} = \frac{u}{2} \begin{bmatrix} \Delta h_1 & \Delta h_1 & 0 & 0 \\ 0 & \ddots & \ddots & 0 \\ 0 & 0 & \Delta h_{N-1} & \Delta h_{N-1} \end{bmatrix} \tag{9}$$

and

$$M_{L \times L} = u \begin{bmatrix} \Delta h'_1 & 0 & 0 \\ 0 & \ddots & 0 \\ 0 & 0 & \Delta h'_N \end{bmatrix} \tag{10}$$



with

$$\boldsymbol{h}' = \begin{bmatrix} 1 & 0 & 0 & 0 \\ .5 & .5 & 0 & 0 \\ 0 & \ddots & \ddots & 0 \\ 0 & 0 & .5 & .5 \\ 0 & 0 & 0 & 1 \end{bmatrix} \boldsymbol{h} \tag{11}$$

Here $\boldsymbol{h}$ has been used as a generalization of the vertical coordinate (altitude, pressure, other) with $L$ elements, while $u$ is the relevant unit conversion constant. Note that in contrast with Eq. (9), the inverse of $\boldsymbol{M}$ in Eq. (10) is not under-constrained,
which favours the latter at the small price of the need for an $\boldsymbol{h}'$.

### 4.1.2 Vertical sampling matching

The number of levels (for point-like concentration values) or layers (for vertically integrated column values) and their vertical locations or boundaries have to be identical for two profiles to be quantitatively compared. One can opt for an explicit vertical range matching of the two profiles first, e.g. by vertical clipping of the one or extension, e.g. by use of a climatology, of the
other. The latter can be applied when later profile operations require knowledge of the atmospheric state over its full vertical range (Keppens et al., 2015). When vertical range matching is skipped, vertical sampling or grid matching operations – often called regridding – automatically limit the height range of the input profile grid to its vertical overlap region with the target profile grid.

Several regridding approaches are in use, although their application typically can depend on units and/or the vertical resolu-
tion discrepancy between the input and target grids:

- Straightforward regridding by (linear or other) interpolation only appropriately works when going from a coarser-resolution input grid to a finer-resolution target grid. Although the corresponding interpolation matrix $\boldsymbol{W}$ is not square, it is applied in an identical way as the unit conversion matrix $\boldsymbol{M}$ in Eq. (8):

$$\boldsymbol{x}' = \boldsymbol{W}\boldsymbol{x}, \boldsymbol{S}' = \boldsymbol{W}\boldsymbol{S}\boldsymbol{W}^T, \boldsymbol{A}' = \boldsymbol{W}\boldsymbol{A}\boldsymbol{W}^* \tag{12}$$

As the inverse of a non-square matrix is ill-posed, the definition of the latter depends on the norm one wishes to minimize. Opting for a simple least-squares difference between the input and target profiles yields $\boldsymbol{W}^* = (\boldsymbol{W}^T\boldsymbol{W})^{-1}\boldsymbol{W}^T$ (Rodgers, 2000).

- When the input grid has a finer resolution than the target grid, one can easily invert the problem by constructing an interpolation matrix $\boldsymbol{W}$ for going from the target grid to the input grid, and then applying the regular regridding formulas
for $\boldsymbol{W}' = \boldsymbol{W}^*$. This approach is denoted as pseudo-inverse (linear or other) regridding. When going from a fine to a coarse vertical sampling, this method approximately conserves the atmospheric constituent's mass (vertically integrated amount) over a wavelet-like vertical window function (also see Section 5).





- In practice vertical sampling definitions might change in time, or one might not know beforehand whether the target grid is coarser than the input grid or vice-versa, or both grids may be similar. Calisesi et al. (2005) have therefore proposed to combine both previous methods by first constructing two interpolation matrices, $\boldsymbol{W}_1$ and $\boldsymbol{W}_2$ respectively, going from the input and target grids to a conjoint super-grid that is the resorted union of the input and target grids. As a result, one

can generally apply vertical sampling matrix $\boldsymbol{W}' = \boldsymbol{W}_2^* \boldsymbol{W}_1$ as before.

- One might instead prefer the total vertical column amount to be conserved during the regridding operation. Such mass-conserved regridding is easily achieved for partial column quantities, whether going from finer to coarser resolution or vice-versa. It is sufficient to construct an overlap matrix that contains the fractions of how much each target grid layer is covered by an input grid layer (Langerock et al., 2015). Assuming that the $i$th output grid layer overlaps with the $j$th

input grid layer, the corresponding element of the conversion matrix is the following interpolation factor:

$$
\begin{aligned}
\boldsymbol{W}(i,j) = \Delta h_{in,j}^{-1} [ & \min(\max(h_{out,i}, h_{in,j})) \\
& - \max(\min(h_{out,i}, h_{in,j}))]
\end{aligned}
\tag{13}
$$

with $\Delta h_{in,j}$ the input layer thickness. Note that this expression implicitly makes use of the conjoint super-grid (see previous), allowing broad usage of this approach. If there is no overlap between target layer $i$ and input layer $j$, then $\boldsymbol{W}(i,j)$ equals 0. The coefficients of the conversion matrix therefore satisfy $0 \leq \boldsymbol{W}(i,j) \leq 1$.

- Total mass-conserved regridding of concentration-type quantities defined on vertical levels or as vertical averages, as is often the case in model fields, is somewhat less straightforward. Before being able to apply the conversion matrix as defined in the previous expression, the point-like concentration values of the input profile must be converted to vertically integrated values, and after the subsequent mass-conserved regridding operation a conversion to the initial units is needed. A combination of Eq. (13) with a forward and backward conversion by use of Eq. (8) including $\boldsymbol{M}$ defined by Eq. (10)

(with well-defined inverse) is hence required, although this can be achieved in arbitrary units (i.e. without the need for the unit conversion constant $u$):

$$
\boldsymbol{W}' = \boldsymbol{M}_{out}^* \boldsymbol{W} \boldsymbol{M}_{in}
\tag{14}
$$

Here $\boldsymbol{M}_{in}$ and $\boldsymbol{M}_{out}$ are the conversion matrices for the input and output grids to their layer representations, respectively, and $\boldsymbol{W}$ is the regular mass-conserved regridding matrix of Eq. (13).

## 4.2   Vertical smoothing matching

The vertical correlation of atmospheric measurement or retrieval quantities results from the allocation to neighbouring levels (layers) of concentrations (columns) that in fact are obtained from vertically overlapping probed air masses. Especially for profile retrievals that have more retrieval levels than independent degrees of freedom in the measurement, the vertical smoothing of the spectral measurement information by the retrieval can be large. As any inversion approach of a retrieval outcome's

vertical smoothing results in an ill-posed deconvolution, vertical smoothing matching is ideally achieved by imposing an





estimator of the coarser height-dependent window smoothing function to each level (layer) of the atmospheric state profile with the smaller window smoothing.

The smoothing window estimator can take any custom-defined shape, but in practice typically a box, triangular, or Gaussian-like function is applied. The window function in any case has to be normalized to unity, while the function width determines

the extent of the vertical smoothing effect. This extent is chosen in agreement with the estimated vertical resolution of the coarser-smoothed atmospheric observation, usually going from a few to several tens of kilometres (Keppens et al., 2015). The smoothing functions additionally have to be discretized to the number of target profile levels (layers) for application of the vertical smoothing matching by matrix multiplication, $\boldsymbol{x}' = \boldsymbol{V}\boldsymbol{x}$, with the rows of $\boldsymbol{V}$ containing the level-specific smoothing functions. The averaging kernel matrix here does not transform in the same way as a representation matching operation (as in

Eqs. (8) and (12)), but is given by $\boldsymbol{A}' = \boldsymbol{V}\boldsymbol{A}$ as a unilateral smoothing of the AK matrix itself (Rodgers and Connor, 2003). On the other hand, the regular conversion formula for the target profile's covariance matrix still holds true, as the covariance represents a quadratic quantity.

For retrieved atmospheric state profiles, the best and already discretized estimators of the vertical smoothing functions are provided by the averaging kernel matrix rows (Rodgers, 2000). These vectors are automatically normalized to unity for some

Philips-Tikhonov-type regularization techniques that have $\boldsymbol{x}_p = \boldsymbol{0}$, but are to be explicitly normalized for optimal estimation and other retrievals that have AKM row sums different from one. The resulting unit-sensitivity averaging kernels (i.e. with unit row sums) are denoted as $\boldsymbol{A}^1$. Usually vertical sampling matching, either of the retrieved profile's averaging kernel matrix or of the target state vector, is required before one can apply $\boldsymbol{V} = \boldsymbol{A}^1$ (also see Section 4.5).

### 4.3    Retrieval matching

Attempting to harmonize two atmospheric state products whereby at least one is the result of a retrieval process, one has to consider differences in measurement weights, prior profile shapes, and prior constraints, in terms of prior covariance matrices, between both products. These differences can be (partially) corrected for in two ways. Either one imposes the retrieval effects of one product on the other, or one annihilates the retrieval effects and associated uncertainties from the retrieved product(s) at the cost of vertical resolution. Both options are discussed in the following two subsections, respectively.

#### 4.3.1    Imposing retrieval effects

     – *Measurement weight matching*: The vertical sensitivity of an atmospheric state retrieval is defined as the column vector of its averaging kernel row sums. It is given by $\boldsymbol{A}\boldsymbol{u}$ if $\boldsymbol{u}$ represents the vertical unit vector and can be considered an estimator of the height-dependent fraction of the retrieval that comes from the measurement, rather than from the prior profile (Rodgers, 2000). One can thus define a diagonal measurement weight matrix $\boldsymbol{W}^M$ (or prior weight matrix

$\boldsymbol{I} - \boldsymbol{W}^M$) by $\mathrm{diag}(\boldsymbol{W}^M) = \boldsymbol{A}\boldsymbol{u}$ so that $\boldsymbol{A} = \boldsymbol{W}^M\boldsymbol{A}^1$ or $\boldsymbol{A}^1 = (\boldsymbol{W}^M)^{-1}\boldsymbol{A}$. The last expression provides the most straightforward calculation of the AK-based vertical smoothing matrix $\boldsymbol{V}$ in the previous section. The measurement weight harmonization operation that matches the sensitivity of an atmospheric state with the measurement weight $\boldsymbol{W}^M$



of a given retrieved state is thus given by $\boldsymbol{x}' = \boldsymbol{W}^M \boldsymbol{x}$, with $\boldsymbol{A}' = \boldsymbol{W}^M \boldsymbol{A}$. Here $\boldsymbol{S}' = \boldsymbol{S}$, as the diagonal matrix $\boldsymbol{W}^M$ can be considered being merely a vertically resolved conversion constant.

– *Prior matching*: Rodgers (2000, Eq. (10.48)) provides an expression for replacing the prior constraint $\boldsymbol{R}_p$ and profile shape $\boldsymbol{x}_p$ within a given retrieval by $\boldsymbol{R}'_p$ and $\boldsymbol{x}'_p$ respectively:

$$\boldsymbol{x}' = (\boldsymbol{S}^{-1} - \boldsymbol{R}_p + \boldsymbol{R}'_p)^{-1}(\boldsymbol{S}^{-1}\boldsymbol{x} - \boldsymbol{R}_p\boldsymbol{x}_p + \boldsymbol{R}'_p\boldsymbol{x}'_p) \tag{15}$$

whereby the prior constraint is typically, but not necessarily, given by the inverse of the prior covariance matrix: $\boldsymbol{R}_p = \boldsymbol{S}_p^{-1}$. If only the prior's profile shape $\boldsymbol{x}_p$ is substituted by $\boldsymbol{x}'_p$ (i.e. for $\boldsymbol{R}'_p = \boldsymbol{R}_p$), the prior matching formula simplifies to the rather intuitive (Rodgers and Connor, 2003, Eq. (10)):

$$\boldsymbol{x}' = \boldsymbol{x} - (\boldsymbol{I} - \boldsymbol{A})(\boldsymbol{x}_p - \boldsymbol{x}'_p) \tag{16}$$

by taking into account that $\boldsymbol{I} - \boldsymbol{A} = \boldsymbol{S}\boldsymbol{S}_p^{-1}$ (Rodgers, 2000, Eq. (2.79)). The latter moreover shows that $\boldsymbol{A}' = \boldsymbol{A} + \boldsymbol{S}\boldsymbol{S}_p^{-1} - \boldsymbol{S}'\boldsymbol{S}_p'^{-1}$. The prior-changed covariance matrix $\boldsymbol{S}'$ is obtained by substituting $\boldsymbol{S}_p$ by $\boldsymbol{S}'_p$ in the retrieval's expression for $\boldsymbol{S}$ (see e.g. Rodgers, 2000, Eq. (2.27)): $\boldsymbol{S}' = (\boldsymbol{S}^{-1} - \boldsymbol{S}_p^{-1} + \boldsymbol{S}_p'^{-1})^{-1}$.

– *Re-optimized prior matching*: By changing the prior in a given retrieval, the resulting atmospheric state profile $\boldsymbol{x}'$ and its AKM will no longer provide an optimally estimated (i.e. with minimal retrieval gain function) representation with respect to the new constraint. Re-optimization of the prior-matched profile might hence be required. When $\boldsymbol{S}'_p = \boldsymbol{S}_p$, this can be achieved by (Rodgers and Connor, 2003, Eq. (18)):

$$\boldsymbol{x}' = \boldsymbol{x}'_p + \boldsymbol{S}'_p\boldsymbol{A}^T(\boldsymbol{A}\boldsymbol{S}'_p\boldsymbol{A}^T + \boldsymbol{S})^{-1}(\boldsymbol{x} - \boldsymbol{x}'_p) \tag{17}$$

whereby the atmospheric state vector $\boldsymbol{x}$ on the right hand side is taken from the output of the prior matching operation in Eq. (16). The re-optimized prior matching that combines Eqs. (16) and (17) thus takes the form $\boldsymbol{x}' = \boldsymbol{P}[\boldsymbol{x} - (\boldsymbol{I} - \boldsymbol{A})\boldsymbol{x}_p] + (\boldsymbol{I} - \boldsymbol{A}')\boldsymbol{x}'_p$, with $\boldsymbol{P} = \boldsymbol{S}'_p\boldsymbol{A}^T(\boldsymbol{A}\boldsymbol{S}'_p\boldsymbol{A}^T + \boldsymbol{S})^{-1}$ and $\boldsymbol{A}' = \boldsymbol{P}\boldsymbol{A}$ like a vertical smoothing operation. Just as before, $\boldsymbol{S}' = \boldsymbol{P}\boldsymbol{S}\boldsymbol{P}^T$ correspondingly. Ridolfi et al. (2006, Eq. (8)) obtained the same conversion matrix $\boldsymbol{P}$ by constructing an optimal interpolation method, i.e. an optimization through $\mathrm{trace}(\boldsymbol{S}_\Delta)$ minimization of combined vertical sampling and smoothing matching operations (hence the non-square AKMs and preceding $\boldsymbol{A}_1$ in their expression). However, based on the complete data fusion framework, Ceccherini et al. have been able to construct a more general re-optimized prior matching operation that is valid for all new prior profile shapes and constraints (Ceccherini et al., 2014, Eq. (7)):

$$\boldsymbol{x}' = \boldsymbol{P}[\boldsymbol{x} - (\boldsymbol{I} - \boldsymbol{A})\boldsymbol{x}_p] + \boldsymbol{P}\boldsymbol{S}(\boldsymbol{A}^T)^{-1}\boldsymbol{R}'_p\boldsymbol{x}'_p \tag{18}$$

This expression even holds when $\boldsymbol{R}'_p$ and $\boldsymbol{x}'_p$ are defined on a different vertical grid than the input profile $\boldsymbol{x}$. In that case it is sufficient to replace $\boldsymbol{A}$ by $\boldsymbol{A}\boldsymbol{W}^*$ in Eq. (18) (also in $\boldsymbol{P}$) with $\boldsymbol{W}$ a regridding matrix as defined in Section 4.1.2 (Ceccherini et al., 2018).





- *Averaging kernel smoothing*: In practice the covariance matrices that are needed in Eqs. (17) and (18) are not always provided to data users, or implementation of the (re-optimized) prior matching is not preferred. One can however avoid these operations by equalling $x'_p$ to the prior of one of the profiles in a comparison, and then applying a vertical smoothing matching and measurement weight matching on this profile by use of the second profile's averaging kernel matrix. In

doing so, only the second profile has to be prior-corrected, resulting in only one non-optimal representation, while this non-optimality and the initial prior constraint of the second profile are enforced on the first profile and therefore drop out of the difference comparison. This whole process thus combines vertical smoothing matching with $V = A^1$, measurement weight matching with $\mathrm{diag}(W^M) = Au$, and prior matching that does not require re-optimization (Eq. (16)). By e.g. comparing a vertically smoothed and measurement weight-corrected reference profile $x'_r = W^M_s A^1_s x_r =$

$A_s x_r$ with a prior shape corrected profile under study $x'_s = x_s - (I - A_s)(x_{p,s} - x_{p,r})$ one obtains (omitting any necessary vertical representation matching):

$$\Delta x = x'_s - x'_r$$
$$= x_s - (I - A_s)(x_{p,s} - x_{p,r}) - W^M_s A^1_s x_r$$
$$= x_s - [A_s x_r + (I - A_s)(x_{p,s} - x_{p,r})] \tag{19}$$

If additionally the reference profile results from an in-situ measurement or model ($x_{p,r} = 0$), this equation can just as well be inferred by considering $x'_s = x_s$ and imposing the satellite retrieval on the reference profile, meaning that the

unknown true profile $x_t$ is replaced by $x_r$ in Eq. (4) (without the error term):

$$x'_r = A_s x_r + (I - A_s)x_{p,s} \tag{20}$$

It is typically the latter interpretation that is referred to as averaging kernel smoothing (of $x_r$). The term however also applies when this reference profile is a retrieved product as well. In that case one can even apply symmetrical smoothing of both the satellite and reference profiles if they show comparable vertical smoothing (Rodgers and Connor, 2003; von

Clarmann and Grabowski, 2007):

$$\Delta x = A_r[x_s - (I - A_s)(x_{p,s} - x_{p,c})]$$
$$- A_s[x_r - (I - A_r)(x_{p,r} - x_{p,c})] \tag{21}$$

for prior matching to a common $x_{p,c}$. This expression simplifies to Eq. (19) for $x_{p,c} = x_{p,r}$ and $A_r = I$.

### 4.3.2 Removing retrieval effects

- *Maximum likelihood representation*: The maximum likelihood representation (MLR) of a retrieved atmospheric state

profile corresponds to the retrieval in the absence of explicit prior information, i.e. the retrieval for $R_p = 0$ [Rodgers, 2000]. One can thus easily convert a given retrieved profile to its maximum likelihood representation by performing a prior matching operation as in Eq. (15) with $R'_p = 0$ (Rodgers, 2000; von Clarmann et al., 2015):

$$x' = (S^{-1} - R_p)^{-1}(S^{-1}x - R_p x_p) \tag{22}$$





The resulting covariance matrix is given by $\boldsymbol{S}' = (\boldsymbol{S}^{-1} - \boldsymbol{S}_p^{-1})^{-1}$, while the averaging kernel matrix becomes the unit matrix, making re-optimization meaningless. This does however not mean that the MLR is fully unconstrained, as it is still implicitly constrained by its vertical grid and the related interpolation convention (von Clarmann and Grabowski, 2007).

5     – *Information-centered representation*: In order to explicitly remove all prior information from a given retrieval and hence simulate a direct measurement with all levels or layers representing one degree of freedom, the prior constraint replacement operation has to be combined with a vertical regridding operation, while also setting $\boldsymbol{x}_p' = \boldsymbol{0}$ (von Clarmann and Grabowski, 2007):

$$\boldsymbol{x}' = \boldsymbol{W}(\boldsymbol{S}^{-1} - \boldsymbol{R}_p + \boldsymbol{R}_p')^{-1}(\boldsymbol{S}^{-1}\boldsymbol{x} - \boldsymbol{R}_p\boldsymbol{x}_p) \tag{23}$$

10     By insertion of the least-squares definition of the pseudo-inverse regridding matrix, resulting in $\boldsymbol{W} = \boldsymbol{W}^{**} = (\boldsymbol{W}^{*T}\boldsymbol{W}^*)^{-1}\boldsymbol{W}^{*T}$, one obtains:

$$\begin{aligned}
\boldsymbol{x}' = (\boldsymbol{W}^{*T}\boldsymbol{S}^{-1}\boldsymbol{W}^* - \boldsymbol{W}^{*T}\boldsymbol{R}_p\boldsymbol{W}^* \\
+ \boldsymbol{W}^{*T}\boldsymbol{R}_p'\boldsymbol{W}^*)^{-1}\boldsymbol{W}^{*T}(\boldsymbol{S}^{-1}\boldsymbol{x} - \boldsymbol{R}_p\boldsymbol{x}_p)
\end{aligned} \tag{24}$$

In order to remove all prior information from the retrieval outcome one thus has to determine $\boldsymbol{W}$ and $\boldsymbol{R}_p'$ that impose the hard constraint $\boldsymbol{W}^{*T}\boldsymbol{R}_p'\boldsymbol{W}^* = \boldsymbol{0}$ non-trivially instead of using the soft MLR constraint $\boldsymbol{R}_p' = \boldsymbol{0}$ (von Clarmann 15   and Grabowski, 2007). The difficulty of this approach lies in the determination of these two matrices in agreement with (i.e. causing minimal loss) the number of independent pieces of information or degrees of freedom in the initial measurement, which is given by $\mathrm{trace}(\boldsymbol{A})$. von Clarmann and Grabowski (2007) provide methods to do so – in both staircase and triangular representation – that are rather extensive and therefore not reproduced here. Eq. (24) can also be obtained from Eq. (18) including $\boldsymbol{W}^*$, while it is in agreement with Rodgers (2000, Eq. (10.50)) only if the latter's 20   back-transformation to the original grid is omitted. Rodgers therefore still indicates his representation as a maximum-likelihood solution, while here the term information-centred representation by von Clarmann and Grabowski (2007) is adopted. Note however that these two references have considered opposite directions in the definition of their respective regridding matrices $\boldsymbol{W}$. Again $\boldsymbol{A}' = \boldsymbol{I}$, while the covariance matrix is now given by $\boldsymbol{S}' = \boldsymbol{W}(\boldsymbol{S}^{-1} - \boldsymbol{S}_p^{-1})^{-1}\boldsymbol{W}^T$ in agreement with the prior matching expression upon addition of a regridding operation.

## 25   4.4   Spatiotemporal co-location matching

As described in the previous sections, vertical sampling and effective resolution differences can be virtually eliminated by applying appropriate regridding and smoothing procedures, respectively. The underlying requirement however is that the vertical dimension within the measurement range is nearly continuously sampled or, phrased differently, that neither the study nor the reference profile is vertically highly under-sampled. This ensures that neither instrument is blind to significantly variable parts 30   of the profile, as only then can interpolation errors be kept to a minimum. Alternatively, interpolation difference errors could be small if both instruments have the same under-sampling pattern, but this hardly occurs in practice.



In the horizontal and temporal dimensions, the sufficient-sampling requirement is usually far from satisfied for vertically resolved atmospheric state observations, in particular for ground-based measurements. Except for some specific measurement campaigns, station-to-station distances are usually much larger than the horizontal representativeness of the measurements, and the typical sounding frequency (e.g. weekly) much coarser than the characteristic measurement duration (minutes to hours) and

time scale of atmospheric variability (Nappo et al., 1982). Consequently, it is usually impossible to horizontally smooth data from multiple ground-based reference stations to the resolution of the measurement under study, just as it does not make sense to interpolate temporally between e.g. weekly soundings. On the other hand, horizontal smoothing can occasionally be achieved for satellite to satellite comparisons that have horizontal averaging kernels available (Lambert et al., 2013). Without the possibility to regrid to a common horizontal and/or temporal grid, comparisons must be done for co-located pairs, whereby

the co-location criteria are designed to ensure minimal co-location mismatch errors in Eq. (5), i.e. minimal differences in the measurements due to a different horizontal and temporal sampling and smoothing of the variable and inhomogeneous atmosphere.

It is beyond the scope of this work to provide a review of all potential co-location methods, which range from simple space and time constraints to more geophysical constraints (e.g. based on potential vorticity), and even Lagrangian trajectory

calculations to match as much as possible the measured air masses (Loew et al., 2017). In this context, it is important to realize that the actual four-dimensional extent of the measurement sensitivity is not easily captured in the metadata (approximations such as an effective measurement location are often too crude). Instead, so-called observation operators can be used to improve the air mass matching (Lambert et al., 2013; Verhoelst et al., 2015). These geometric parametrizations of the four-dimensional extent of the measurement sensitivity are based on physical considerations and – if possible – radiative transfer and retrieval

models. They can for instance be derived from dedicated calculations of horizontal averaging kernels (von Clarmann et al., 2009).

Despite these attempts to optimize the co-location criteria, some irreducible co-location mismatch usually still affects the comparisons, adding non-negligible random and systematic errors to the difference statistics, and thereby hampering the interpretation of the differences in terms of the quality of the measurements and their reported uncertainties. Several approaches

to quantify these co-location difference errors exist, e.g. see Verhoelst et al. (2015) and Fassó et al. (2017) for an overview and some case studies. Particularly appealing is the option to estimate the individual errors from model-based simulations. In this approach, the measurements are simulated by applying the observation operators, initialized with the real measurement metadata, on a gridded representation of the atmosphere. The vertically resolved difference $\Delta m$ between the simulated measurement under study $m_s$ and the simulated reference measurement $m_r$ then provides an estimate of the horizontal and

temporal co-location mismatch error profile:

$$\Delta m = \langle \epsilon_{\Delta Hsa} + \epsilon_{\Delta Tsa} + \epsilon_{\Delta Hsm} + \epsilon_{\Delta Tsm} \rangle \tag{25}$$

This co-location mismatch error estimate can be used to horizontally and temporally match the observed profiles (von Clarmann, 2006, Eq. (15)):

$$x' = x - \Delta m \tag{26}$$




The use of model data however also introduces some model uncertainty in the comparison results, meaning that this procedure only makes sense when the model uncertainty is (expected to be) smaller than the (spread on the) co-location mismatch errors. Moreover, a residual co-location difference error is still present, caused by finer structures in the sampling and smoothing of the observations than those accounted for by the model. This residual error can be quantified by use of an additional
reference dataset that has a finer resolution than the model (von Clarmann, 2006), but the quantification procedure is not expanded here. Combining the model uncertainty and (possibly negligible) residual spatiotemporal co-location difference error into $S_{\Delta m}$, one simply has $S' = S + S_{\Delta m}$. Although strictly speaking the averaging kernel matrix is no longer valid for the spatiotemporally shifted profile $x'$, one can estimate the effect of the model uncertainty that is introduced during the matching operation on the AKM by taking $A' = I - S' S_p^{-1} = I - (S + S_{\Delta m}) S_p^{-1} = A - S_{\Delta m} S_p^{-1}$.

## 4.5  Overview and order of operations

An overview of the atmospheric state profile matching operations discussed in this work is listed in Table 1 (order of appearance). The matrix algebra that is required to obtain $x'$, $S'$, and $A'$ is provided for each operation. The flowchart in Fig. 1 on the other hand shows the preferred order of the matching operations that possibly precede the comparison of two atmospheric state profiles under study. Vertical representation matching (of quantities and grids) is thereby mandatory, but the full annihilation
of retrieval effects by changing to the information-centered representation has to take place first, as it also includes a change in the profile's vertical sampling. Optional vertical smoothing matching, measurement weight matching, and prior matching follow after the representation matching. All three can be combined into the so-called averaging-kernel smoothing operation, or one can opt for a conversion to a maximum-likelihood representation for one or both profiles. Both options do not require re-optimization operations.

Keppens et al. (2015) have discussed the possibility to perform averaging kernel smoothing by multiplying a row-interpolated averaging kernel matrix with a full high-resolution ground profile, instead of regridding the ground profile first as suggested in Fig. 1. The former approach maximally exploits the fine-gridded reference measurement without adding information to the retrieval data (Ridolfi et al., 2006). On the other hand however, this method additionally requires row-renormalization of the interpolated AKM in order to conserve the vertical sensitivity of the averaging kernel matrix (Keppens et al., 2015, Eq. (11)).
Only for mass-conserved regridding of partial column quantities is the AKM renormalization already included by definition. In that case both approaches are equivalent, as one has $A'x = Ax' = AWx$. Keeping all vertical sampling matching operations before any averaging kernel smoothing therefore in general is the most straightforward approach. This order of operations moreover avoids the smoothing error pitfalls as discussed by von Clarmann (2014).

## 5  Harmonization impact

While intended to merely remove uncertainty contributions from eventual atmospheric state profile difference statistics, the harmonization operations discussed in this work obviously also impact the remaining covariance (matrix) and the information that is contained within a retrieval's averaging kernel matrix. First of all, from the discussion on vertical smoothing matching





**Table 1.** Matching operations overview table for vertically resolved atmospheric state observations (order of appearance in the text). The averaging kernel (AK) smoothing operation shows the exemplary case for $\boldsymbol{x}_{p,c} = \boldsymbol{x}_{p,r} = \boldsymbol{0}$ and $\boldsymbol{A}_r = \boldsymbol{I}$.

| Matching operation | $\boldsymbol{x'}$ | $\boldsymbol{S'}$ | $\boldsymbol{A'}$ |
|---|---|---|---|
| Vert. quantity matching | $\boldsymbol{Mx}$ | $\boldsymbol{MSM}^T$ | $\boldsymbol{MAM}^{-1}$ |
| Vert. sampling matching | $\boldsymbol{Wx}$ | $\boldsymbol{WSW}^T$ | $\boldsymbol{WAW}^*$ |
| Vert. smoothing matching | $\boldsymbol{Vx}$ | $\boldsymbol{VSV}^T$ | $\boldsymbol{VA}$ |
| Meas. weight matching | $\boldsymbol{W}^M\boldsymbol{x}$ | $\boldsymbol{S}$ | $\boldsymbol{W}^M\boldsymbol{A}$ |
| Prior matching (PM) | $\boldsymbol{S'}(\boldsymbol{S}^{-1}\boldsymbol{x} - \boldsymbol{R}_p\boldsymbol{x}_p + \boldsymbol{R}'_p\boldsymbol{x}'_p)$ | $(\boldsymbol{S}^{-1} - \boldsymbol{R}_p + \boldsymbol{R}'_p)^{-1}$ | $\boldsymbol{A} + \boldsymbol{SS}_p^{-1} - \boldsymbol{S'S}'^{-1}_p$ |
| Re-optimized PM | $\boldsymbol{P}[\boldsymbol{x} - (\boldsymbol{I} - \boldsymbol{A})\boldsymbol{x}_p] + \boldsymbol{PS}(\boldsymbol{A}^T)^{-1}\boldsymbol{R}'_p\boldsymbol{x}'_p$ | $\boldsymbol{PSP}^T$ | $\boldsymbol{PA}$ |
| AK smoothing (for $s$ on $r$) | $\boldsymbol{A}_s\boldsymbol{x}_r + (\boldsymbol{I} - \boldsymbol{A}_s)\boldsymbol{x}_{p,s}$ | $\boldsymbol{A}_s^1\boldsymbol{S}_r(\boldsymbol{A}_s^1)^T$ | $\boldsymbol{A}_s$ |
| Maximum likelihood repr. | $\boldsymbol{S'}(\boldsymbol{S}^{-1}\boldsymbol{x} - \boldsymbol{R}_p\boldsymbol{x}_p)$ | $(\boldsymbol{S}^{-1} - \boldsymbol{R}_p)^{-1}$ | $\boldsymbol{I}$ |
| Information-centered repr. | $\boldsymbol{W}(\boldsymbol{S}^{-1} - \boldsymbol{R}_p)^{-1}(\boldsymbol{S}^{-1}\boldsymbol{x} - \boldsymbol{R}_p\boldsymbol{x}_p)$ | $\boldsymbol{W}(\boldsymbol{S}^{-1} - \boldsymbol{R}_p)^{-1}\boldsymbol{W}^T$ | $\boldsymbol{I}$ |
| Co-location matching | $\boldsymbol{x} - \boldsymbol{\Delta m}$ | $\boldsymbol{S} + \boldsymbol{S}_{\Delta m}$ | $\boldsymbol{A} - \boldsymbol{S}_{\Delta m}\boldsymbol{S}_p^{-1}$ |

one can observe that in fact all operations that include a multiplication with a non-diagonal conversion matrix also impose a vertical smoothing on the vertical profile and its covariance and averaging kernel matrices. Especially the vertical sampling matching operation combines information from several input grid levels into a single output grid level by definition. For linear and mass-conserved regridding operations the associated vertical smoothing windows are approximately triangular and

square respectively, with an extent that is limited to adjacent grid points (see Fig. 2). When going from a fine to a coarse grid however, the use of inverse or double (linear) interpolation over a conjoint super-grid results in a wavelet-shaped vertical smoothing function that can extend up to the full vertical profile range. This is due the pseudo-inverse matrix that is involved, as demonstrated in Fig. 2.

   Vertical quantity matching by use of a diagonal conversion matrix will not introduce a vertical smoothing effect, but affects

the covariance matrix and the averaging kernel matrix nevertheless. This is a result of these matrices being typically provided in absolute and thus unit-dependent numbers. One can avoid this unit-dependence by switching to fractional representations of the covariance and averaging kernel matrices instead. These are given by $\boldsymbol{S}_R(i,j) = \boldsymbol{S}(i,j)\boldsymbol{x}(i)^{-1}\boldsymbol{x}(j)^{-1}$ and $\boldsymbol{A}_R(i,j) = \boldsymbol{A}(i,j)\boldsymbol{x}(i)^{-1}\boldsymbol{x}(j)$ respectively (Keppens et al., 2015, Eqs. (3) and (4)). Note that the latter automatically results from a logarithmic retrieval. Because of their invariance under (matrix-diagonal) unit conversions, such fractional averaging kernel

matrices are preferred for information content studies (Keppens et al., 2015). Fractional kernel representations are therefore also assumed in Table 2 that summarizes how a retrieval's degrees of freedom in the signal (DFS), calculated as the AKM trace, and its vertical sensitivity, calculated as the AKM row sum vector, are altered by each harmonization operation.




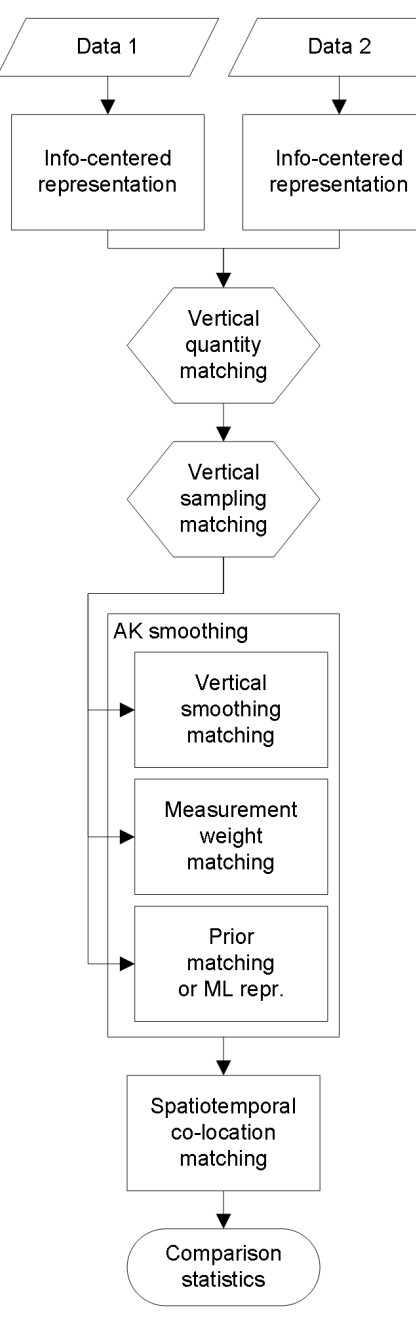

**Figure 1.** Profile harmonization flowchart, indicating the order of the matching operations outlined in the text. Rectangular boxes are optional, while hexagons are mandatory. The maximum likelihood (ML) representation has here been included as a prior matching operation with $R'_p = 0$.



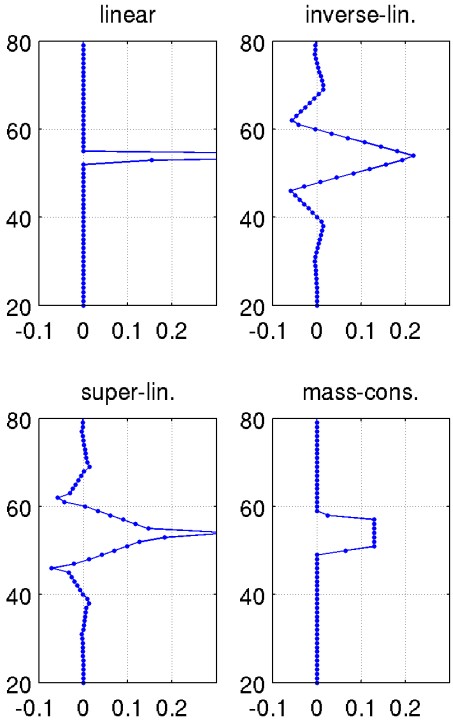

**Figure 2.** Regridding window functions for the four vertical sampling matching operations discussed in Section 4.1.2, going from a fine grid (0 to 100 in steps of one) to a coarse grid (0 to 100 in steps of 100/13) that only overlaps at the vertical edges (a.u.). The seventh row of each regridding matrix with dimension $14 \times 101$ is plotted (columns 20 to 80 only). For pseudo-inverse and super-linear regridding these matrix elements are never zero, but of the order of $10^{-6}$ at the edges.

## 6 Uncertainty budget

The harmonization operations presented in this work are intended to enable the calculation of profile difference statistics and to eliminate uncertainty contributions from the total uncertainty budget as expressed by Eq. (7). Table 3 lists for each profile matching operation the covariance that is thereby removed (first column), how the ex-ante covariance of the harmonized atmospheric state product is altered (second column), and what uncertainty is possibly introduced by the operation or remains as a residual despite the matching (third column).

It is clear that the vertical representation harmonization operations do actually not remove uncertainty from the full budget, but are required for difference calculations of atmospheric state vectors with equal units and lengths. These operations affect the product covariance and moreover introduce auxiliary unit-conversion quantity uncertainty $S_Q$ and an additional vertical smoothing difference uncertainty $S'_{\Delta V sm}$ (see regridding impact discussion in previous section and next paragraph), respectively. The former however is usually hard to quantify, and therefore often neglected. The model uncertainty $S_{\Delta m}$ that is





**Table 2.** Impact of matching operations on the information that is contained in the fractional averaging kernel matrix, as expressed by the DFS = $\mathrm{trace}(\boldsymbol{A})$ and the vertical sensitivity $\boldsymbol{Au}$ (order of appearance in the text). The averaging kernel (AK) smoothing operation shows the exemplary case for $\boldsymbol{x}_{p,c} = \boldsymbol{x}_{p,r} = \boldsymbol{0}$ and $\boldsymbol{A}_r = \boldsymbol{I}$.

| Matching operation | DFS = $\mathrm{trace}(\boldsymbol{A}')$ | Vertical sensitivity = $\boldsymbol{A}'\boldsymbol{u}$ |
| --- | --- | --- |
| Vert. quantity matching | $\mathrm{trace}(\boldsymbol{A})$ | $\boldsymbol{Au}$ |
| Vert. sampling matching | $\mathrm{trace}(\boldsymbol{WAW}^*)$ | $\boldsymbol{WAW}^*\boldsymbol{u}$ |
| Vert. smoothing matching | $\mathrm{trace}(\boldsymbol{VA})$ | $\boldsymbol{VAu}$ |
| Meas. weight matching | $\mathrm{trace}(\boldsymbol{W}^M\boldsymbol{A})$ | $\boldsymbol{W}^M\boldsymbol{Au}$ |
| Prior matching (PM) | $\mathrm{trace}(\boldsymbol{A}) + \mathrm{trace}(\boldsymbol{SS}_p^{-1}) - \mathrm{trace}(\boldsymbol{S}'\boldsymbol{S}_p'^{-1})$ | $\boldsymbol{Au} + \boldsymbol{SS}_p^{-1}\boldsymbol{u} - \boldsymbol{S}'\boldsymbol{S}_p'^{-1}\boldsymbol{u}$ |
| Re-optimized PM | $\mathrm{trace}(\boldsymbol{S}_p'\boldsymbol{A}^T(\boldsymbol{AS}_p'\boldsymbol{A}^T + \boldsymbol{S})^{-1}\boldsymbol{A})$ | $\boldsymbol{S}_p'\boldsymbol{A}^T(\boldsymbol{AS}_p'\boldsymbol{A}^T + \boldsymbol{S})^{-1}\boldsymbol{Au}$ |
| AK smoothing (for $s$ on $r$) | $\mathrm{trace}(\boldsymbol{A}_s)$ | $\boldsymbol{A}_s\boldsymbol{u}$ |
| Maximum likelihood repr. | $\mathrm{rank}(\boldsymbol{A})$ | $\boldsymbol{u}$ |
| Information-centered repr. | $\mathrm{rank}(\boldsymbol{A}')$ | $\boldsymbol{u}'$ |
| Co-location matching | $\mathrm{trace}(\boldsymbol{A}) - \mathrm{trace}(\boldsymbol{S}_{\Delta m}\boldsymbol{S}_p^{-1})$ | $\boldsymbol{Au} - \boldsymbol{S}_{\Delta m}\boldsymbol{S}_p^{-1}\boldsymbol{u}$ |

introduced by the co-location matching operation (see Section 4.4) is of the same nature as $\boldsymbol{S}_Q$, but preferably better characterized and explicitly taken into account as the model correction of a vertical profile leaves the associated ex-ante product uncertainty unchanged: $\boldsymbol{S}' = \boldsymbol{S} + \boldsymbol{S}_{\Delta m}$. Note that despite these additional uncertainties one evidently expects the matching operations to reduce the overall difference covariance. For sufficiently fine-gridded models the co-location matching could in
principle also account for vertical sampling and smoothing differences, but this is hardly feasible in practice.

Two atmospheric state products with different vertical smoothing $\boldsymbol{V}_1$ and $\boldsymbol{V}_2$ respectively have a vertical smoothing difference covariance $\boldsymbol{S}_{\Delta Vsm} = (\boldsymbol{V}_1 - \boldsymbol{V}_2)\boldsymbol{S}_C(\boldsymbol{V}_1 - \boldsymbol{V}_2)^T$ in their combined uncertainty budget (e.g. Rodgers and Connor, 2003; von Clarmann and Grabowski, 2007). $\boldsymbol{S}_C$ here represents the comparison ensemble's covariance matrix, which in practice is often replaced by one of the two ex-ante product covariance matrices or their sum. Upon vertical smoothing matching, e.g. by
10 enforcing the vertical smoothing of the first on the second, the vertical smoothing difference is actually not fully removed, as a residual smoothing difference covariance $\boldsymbol{S}'_{\Delta Vsm}$ remains:

$$\boldsymbol{S}'_{\Delta Vsm} = (\boldsymbol{V}_1 - \boldsymbol{V}_1\boldsymbol{V}_2)\boldsymbol{S}_C(\boldsymbol{V}_1 - \boldsymbol{V}_1\boldsymbol{V}_2)^T \tag{27}$$

(or $\boldsymbol{S}'_{\Delta Vsm} = (\boldsymbol{V}_2\boldsymbol{V}_1 - \boldsymbol{V}_1\boldsymbol{V}_2)\boldsymbol{S}_C(\boldsymbol{V}_2\boldsymbol{V}_1 - \boldsymbol{V}_1\boldsymbol{V}_2)^T$ for symmetrical smoothing). Hence only the vertical smoothing of an ideal measurement with $\boldsymbol{V}_2 = \boldsymbol{I}$ fully eliminates the vertical smoothing difference error (von Clarmann and Grabowski,
2007). It is the latter case that typically occurs for the vertical smoothing of model data and in-situ reference data. When also considering vertical sampling matching, e.g. of the second product, $\boldsymbol{V}_2$ has to be replaced by $\boldsymbol{W}\boldsymbol{V}_2\boldsymbol{W}^*$ in Eq. (27) (von Clarmann, 2014).

For the (asymmetrical) averaging kernel smoothing operation the expression in Eq. (27) has been modified to include the study and reference product AKMs. This harmonization operation also includes a measurement weight matching and a prior




matching (see Section 4.3 and Fig. 1). Only the residual smoothing difference error covariance thus remains. The measurement weight matching actually consists of a rescaling and does therefore not introduce a new covariance term. Non-optimal prior matching on the other hand corrects for differences in prior profile shape and prior constraint, but as a result changes the measurement weight difference covariance to $\boldsymbol{S}'_{\Delta MW}$. Re-optimization of the prior-corrected state by use of Eq. (17) corrects

for this measurement weight difference, yet alters the vertical smoothing difference error as a result.

In terms of the uncertainty contributions that are removed from the full covariance of the difference, the AK smoothing operation is equivalent to the re-optimized prior matching (Eq. (18)) and to switching to the information-centered representation beforehand. While for the former only a residual vertical smoothing difference error defined by $\boldsymbol{P}$ remains, the latter operation changes the vertical sampling difference covariance due to the inherent regridding operation (which upon subsequent vertical

sampling matching is replaced by a vertical smoothing difference error). As demonstrated by von Clarmann and Grabowski (2007, Eq. (58)), the information-centered representation yields an additional residual smoothing difference error if the variability of the true state is not sufficiently well characterized by $\boldsymbol{S}'$ (not assumed here). The maximum likelihood representation aims at removing all prior information (including vertical smoothing and measurement weight), but actually is still implicitly (prior-)constrained by its vertical grid (see Section 4.3). Therefore a residual vertical smoothing difference and prior constraint

difference contribution must be considered in the uncertainty budget.

**Table 3.** Impact of matching operations on the comparison uncertainty budget $\boldsymbol{S}_\Delta$ (order of appearance in the text). The residual covariance of the vertical sampling matching includes the regridded vertical smoothing $\boldsymbol{V}'_2 = \boldsymbol{W}\boldsymbol{V}_2\boldsymbol{W}^*$. The averaging kernel (AK) smoothing operation shows the unidirectional case.

| Matching operation | Removed covariance | Converted covariance $\boldsymbol{S}'$ | Introduced and residual covariance |
|---|---|---|---|
| Vert. quantity matching | / | $\boldsymbol{MSM}^T$ | $\boldsymbol{S}_Q$ |
| Vert. sampling matching | / | $\boldsymbol{WSW}^T$ | $(\boldsymbol{V}_1 - \boldsymbol{V}'_2)\boldsymbol{S}_C(\boldsymbol{V}_1 - \boldsymbol{V}'_2)^T$ |
| Vert. smoothing matching | $\boldsymbol{S}_{\Delta Vsm}$ | $\boldsymbol{VSV}^T$ | $(\boldsymbol{V}_1 - \boldsymbol{V}_1\boldsymbol{V}_2)\boldsymbol{S}_C(\boldsymbol{V}_1 - \boldsymbol{V}_1\boldsymbol{V}_2)^T$ |
| Meas. weight matching | $\boldsymbol{S}_{\Delta MW}$ | $\boldsymbol{S}$ | / |
| Prior matching (PM) | $\boldsymbol{S}_{\Delta PS}$ and/or $\boldsymbol{S}_{\Delta PC}$ | $(\boldsymbol{S}^{-1} - \boldsymbol{R}_p + \boldsymbol{R}'_p)^{-1}$ | $\boldsymbol{S}'_{\Delta MW}$ |
| Re-optimized PM | $\boldsymbol{S}_{\Delta Vsm} + \boldsymbol{S}_{\Delta PS} + \boldsymbol{S}_{\Delta PC} + \boldsymbol{S}_{\Delta MW}$ | $\boldsymbol{PSP}^T$ | $(\boldsymbol{A} - \boldsymbol{PA})\boldsymbol{S}_C(\boldsymbol{A} - \boldsymbol{PA})^T$ |
| AK smoothing (for $s$ on $r$) | $\boldsymbol{S}_{\Delta Vsm} + \boldsymbol{S}_{\Delta PS} + \boldsymbol{S}_{\Delta PC} + \boldsymbol{S}_{\Delta MW}$ | $\boldsymbol{A}^1_s\boldsymbol{S}_r(\boldsymbol{A}^1_s)^T$ | $(\boldsymbol{A}_s - \boldsymbol{A}_s\boldsymbol{A}_r)\boldsymbol{S}_C(\boldsymbol{A}_s - \boldsymbol{A}_s\boldsymbol{A}_r)^T$ |
| Maximum likelihood repr. | $\boldsymbol{S}_{\Delta Vsm} + \boldsymbol{S}_{\Delta PS} + \boldsymbol{S}_{\Delta PC} + \boldsymbol{S}_{\Delta MW}$ | $(\boldsymbol{S}^{-1} - \boldsymbol{R}_p)^{-1}$ | $\boldsymbol{S}'_{\Delta Vsm} + \boldsymbol{S}'_{\Delta PC}$ |
| Information-centered repr. | $\boldsymbol{S}_{\Delta Vsm} + \boldsymbol{S}_{\Delta PS} + \boldsymbol{S}_{\Delta PC} + \boldsymbol{S}_{\Delta MW}$ | $\boldsymbol{W}(\boldsymbol{S}^{-1} - \boldsymbol{R}_p)^{-1}\boldsymbol{W}^T$ | $\boldsymbol{S}'_{\Delta Vsa}$ |
| Co-location matching | $\boldsymbol{S}_{\Delta Hsa} + \boldsymbol{S}_{\Delta Tsa} + \boldsymbol{S}_{\Delta Hsm} + \boldsymbol{S}_{\Delta Tsm}$ | $\boldsymbol{S}$ | $\boldsymbol{S}_{\Delta m}$ |

# 7   Conclusions

In the context of data comparisons as performed in satellite validation, and data combinations through assimilation or fusion, this work discusses the most frequent methods for the harmonization of vertically resolved atmospheric state observations in a




conceptually and terminologically aligned framework. The harmonization of two profiles' representations is required to enable data comparisons and for proper quantitative $\chi^2$ testing of the resulting total difference covariance. Other data manipulations are needed to reduce, in the uncertainty budget of the data comparison, the contributions of differences in retrieval characteristics and of spatiotemporal mismatches. Ten matching operations have been identified and expressed by common matrix algebra.

The latter includes conversion of the ex-ante covariance matrix and the averaging kernel matrix (for retrieved products) that are associated to each atmospheric profile retrieval. It has therefore also been discussed how the information content of a retrieved product, as calculated from its AKM, is modified by each harmonization operation. Finally it has been examined what covariance is removed from the full comparison uncertainty budget by each harmonization operation, and what covariance remains as a residual or is introduced as a result. Concerning the covariance terms removed, averaging kernel smoothing appears

to be equivalent to re-optimized prior matching and to switching to the information-centered representation beforehand, which both however are more difficult to practically implement. These operations only leave a residual smoothing difference error in the comparison (after regridding to a joint vertical grid for the latter). In combination with co-location matching by use of model data these three approaches reduce the difference covariance to its minimum of the form $\boldsymbol{S}'_\Delta = \boldsymbol{S}'_1 + \boldsymbol{S}'_2 + \boldsymbol{S}'_{\Delta V sm} + \boldsymbol{S}_{\Delta m}$.

*Competing interests.* The authors declare that they have no conflict of interest.

*Disclaimer.* Special issue statement. This article is part of the special issue "Towards Unified Error Reporting (TUNER)" edited by T. von Clarmann, D. Degenstein, N. Livesey, and H. Worden (von Clarmann et al., 2019, paper in preparation). It does not belong to a conference.

*Acknowledgements.* The reported work was funded by ESA via the CCI–ECV Ozone project, and by the Belgian Federal Science Policy Office (BELSPO) and ESA via the ProDEx project TROVA (PEA 4000116692, supporting S5PVT AO ID 28587 CHEOPS-5p). This work builds on the versatile satellite validation system Multi-TASTE that was developed in several heritage projects and refined within the EU FP7

Project Quality Assurance for Essential Climate Variables (QA4ECV), grant no. 60740, and EU H2020 project Gap Analysis for Integrated Atmospheric ECV CLImate Monitoring (GAIA-CLIM), grant no. 640276. The authors would like to acknowledge Thomas von Clarmann, Simone Ceccherini, Nicola Zoppetti, and Viktoria Sofieva for helpful discussions.




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
