# Peer review of "Harmonization and comparison of vertically resolved atmospheric state observations: Methods, effects, and uncertainty budget"

_Atmospheric Measurement Techniques, 2019_

## Referee Comment (RC1) · Anonymous Referee #1 · 21 May 2019

Review of "Harmonization and comparison of vertically resolved atmospheric state observations: Methods, effects and uncertainty budget"

by Keppens et al,

GENERAL COMMENTS

The paper, written by a team of authors who have much experience of comparison of various types of satellite data product and reference measurements, represents a worthy attempt to present a unified framework for all such comparisons, while also giving due recognition to earlier work by other authors who have dealt with more specific aspects.

[Figure]

Unlike the authors, most readers will only be familiar with limited applications of the intercomparison problem, so the challenge is to write a paper general enough to cover the whole field, while also making it understandable to those only working on particular aspects. It's not an easy read, but I don't see how it could be done better, and the flow chart and tables offer useful summaries.

Hopefully this paper will serve as the reference document for future discussions of validation methodology.

MINOR COMMENTS

I have only minor suggestions for clarifications and grammar

P4 L6: 'which are' instead of 'who are'

P5 Eqs 9,10: Should subscripts in the bracketed terms be 'L' rather than 'N'?

P7 Eq 13: This equation made no sense to me.

P8, L21: Assuming 'prior constraints in terms of prior covariance matrices' is to be taken as a single entity, I suggest removing the comma after 'constraints' otherwise it may appear that measurement weights and prior profile shapes are also in terms of prior covariance matrices.

P8, L23 (and elsewhere): I suggest 'retrieval artefacts' rather than 'retrieval effects' to emphasise that these are unintended rather than intended consequences of the retrieval process.

P8, L24 (and elsewhere): I suggest 'eliminates' rather than 'annihilates' 'Annihilate' has violent connotations (although perhaps the authors do have particularly strong feelings on this matter?).

P11 Eq 23-24: Getting from Eq 23 to Eq 24 doesn't seem obvious. Does this require some trick along the lines of getting from 4.11 to 4.12 in Rodgers?

---

## Referee Comment (RC2) · Anonymous Referee #2 · 24 May 2019

The manuscript "Harmonization and comparison of vertically resolved atmospheric state observations: methods, effects, and uncertainty budget" by A. Keppens et al. presents a summary of different methods for harmonization and comparison of atmospheric profile retrievals. This paper summarizes different approaches to harmonize data using a consistent mathematical terminology, which makes it easy to contrast different methods and estimate a contribution of each manipulation to the final result.

This study fits well to the scope of the problems discussed in the ATM. This manuscript would be a good addition to the special TUNER (Towards Unified Error Reporting) issue laying a theoretical base for the atmospheric profile comparisons and error analysis.

[Figure]

The results are presented in a consistent manner. The manuscript is very technical, but overall it is well written. I would recommend this paper for the publication in the AMT after some technical corrections. My general and technical comments are listed below.

General comments:

-Page 4, line 5: Do you mean cross-covariances here - the covariances between different error terms? Please, re-phrase this statement.

-Page 6, lines 16-19: I am not sure if I understand this part of the manuscript. To interpolate a vertical profile from a coarser vertical scale to a finer scale, one needs to assume an underlying function (linear, quadratic etc.). This interpolation might introduce an additional source of error into the comparison results that authors did not mentioned and considered here. If one degrades vertical resolution of the profile reported on a finer vertical scale, then no additional information is required. Typically, it is recommended to make comparisons on a vertical scale of the measurement with a coarser resolution. There is a discussion of effect of interpolation in section 5, but it would be good if you can mention that here as well.

-Page 7, equation 13: Why do you apply two operators (max/min) simultaneously? How would these two operators work? In the original paper referenced here [Langerock et al., 2015], authors use min or max, but not a combination of the two.

-Page 19, Conclusions. Conclusions are not well written, and I would suggest you make some revision. I listed some suggestions below in the technical section.

Technical comments:

-Page 1, line 11: the reference is missing;

-Page 2, lines1-2: there are many abbreviations here that have not been introduced earlier in the text.

-Page 3, lines 1-10: Typically, Xa and Sa are used to define apriori profile and corresponding matrix. It might be easier for readers to follow your paper if you use the established terms.

-Page 4, line 18: It might be better to re-phrase: "a full assessment and quantification of all contributions to the difference error Sdelta are necessary . . ."

-Page 5, lines 6-7: I suggest to replace this phrase with ". . .the latter may introduce a bias and increase the spread due to uncertainties in the ancillary data. . . "

-Page 5, eq. 8 (also eq. 12): there are three equations written in one line. I suggest to have one equation in each line and have (a, b, c) labels for each equation.

-Page 7, lines 29-30: the sentence needs some re-wording, because the meaning is not clear.

-Page 19, line 1: I suggest to re-phrase with "The harmonization of a pair of atmospheric profile retrievals and their representations is required . . ."

-Page 19, lines 3-4: This sentence needs some re-wording. Maybe something like "Other data manipulations are needed to reduce the error budget of the comparison by minimizing contributions due to differences in retrieval characteristics and spatiotemporal sampling."

-Page 19, lines 4-5: replace with ". . . have been identified and expressed in a consistent way using common matrix algebra."

-Page 19, lines 5-6: Suggest to replace with "These operations include procedures for converting the axe-ante covariance and averaging kernels matrices associated to each atmospheric profile retrieval."

-Page 19, lines 7-10. It would be easy to read and understand if you simply say "In this study we discussed . . ." or "Finally, we examined . . .".

-Page 19, lines 8-10. I would rather say ". . . what terms of the error covariance are

removed. . ."

---

## Author Comment (AC1) · 18 Jun 2019

The response to the Referees is structured in a clear and easy-to-follow sequence: (1) comments from Referees; (2) author's response; (3) author's changes in manuscript.

GENERAL COMMENTS (1) The paper, written by a team of authors who have much experience of comparison of various types of satellite data product and reference measurements, represents a worthy attempt to present a unified framework for all such comparisons, while also giving due recognition to earlier work by other authors who have dealt with more specific aspects. Unlike the authors, most readers will only be familiar with limited applications of the intercomparison problem, so the challenge is

to write a paper general enough to cover the whole field, while also making it understandable to those only working on particular aspects. It's not an easy read, but I don't see how it could be done better, and the flow chart and tables offer useful summaries. Hopefully this paper will serve as the reference document for future discussions of validation methodology. (2) The authors highly appreciate the reviewer's positive view on the manuscript, and are very grateful for the encouraging words. (3) No changes have been made based on the general comments.

MINOR COMMENTS I have only minor suggestions for clarifications and grammar:

(1) P4 L6: 'which are' instead of 'who are' (2) Agree. (3) This suggestion has been implemented.

(1) P5 Eqs. 9, 10: Should subscripts in the bracketed terms be 'L' rather than 'N'? (2) Thanks for noting this erroneous remnant of a previous notation. (3) Lower indices N have been replaced by L in Eqs. (9) and (10).

(1) P7 Eq. 13: This equation made no sense to me. (2) It is agreed that equation 13 is not fully clear in its present form. In fact, the nested max and min in the two terms indicated the layers' upper and lower bounds respectively, and not the mathematical max or min (as for the first occurrences). This has now been made clear in the equation and the text. (3) The second max and min in each term have been replaced by an upper height index U and L, respectively, and the first sentence following the equation has been extended with the following: "and the upper indices U and L indicating the layers' upper and lower height bounds, respectively"

(1) P8, L21: Assuming 'prior constraints in terms of prior covariance matrices' is to be taken as a single entity, I suggest removing the comma after 'constraints' otherwise it may appear that measurement weights and prior profile shapes are also in terms of prior covariance matrices. (2) The authors agree that the current formulation including the comma is misleading. Removing the comma however suggests that prior constraints are always expressed in terms of prior covariance matrices, which is not

the case. The authors therefore suggest to just remove the text part between the commas, to keep the discussion as general as possible at this point in the manuscript. Later in the text, below Eq. (15), the following text part (already there in the discussion paper) sufficiently specifies what is needed for further discussion: "whereby the prior constraint is typically, but not necessarily, given by the inverse of the prior covariance matrix" (3) The text part "in terms of prior covariance matrices" has been removed from the first sentence of Section 4.3.

(1) P8, L23 (and elsewhere): I suggest 'retrieval artefacts' rather than 'retrieval effects' to emphasise that these are unintended rather than intended consequences of the retrieval process. (2) The authors agree, also in order not the confuse with the 'effects' of the harmonization operations. (3) Occurrences in the text of "retrieval effects" have been replaced by "retrieval artefacts".

(1) P8, L24 (and elsewhere): I suggest 'eliminates' rather than 'annihilates' 'Annihilate' has violent connotations (although perhaps the authors do have particularly strong feelings on this matter?). (2) Although "annihilation" has appeared in the literature before, it is not a fixed or generally applied expression. The authors therefore agree with the reviewer's suggestion. (3) Occurrences in the text of "annihilation" or variations thereon have been replaced by "elimination" or variations thereon.

(1) P11 Eq. 23-24: Getting from Eq. 23 to Eq. 24 doesn't seem obvious. Does this require some trick along the lines of getting from 4.11 to 4.12 in Rodgers? (2) In order to obtain Eq. (24), it is sufficient to apply the matrix algebra property $(AB)^* = B^*A^*$ multiple times, indeed as in Rodgers (2000) 4.11 to 4.12, or 10.49 to 10.50, as referred to in the paragraph below Eq. (24). The authors however agree that the current suggestion of inserting the least squares definition of the pseudo-inverse regridding matrix is not fully straightforward. Eq. (24) is obtained more directly if one neutrally inserts the transposed regridding matrix and its pseudo-inverse directly. This suggestion has now been adopted in the text, and Eq. (24) has been extended with an intermediate step. (3) The sentence between Eqs. (23) and (24) has been rewritten: "By insertion of the

transposed regridding matrix and its pseudo-inverse, one obtains:" Eq. (24) has been extended to explicitly show this insertion as an intermediate calculation step.

---

## Author Comment (AC2) · 18 Jun 2019

The response to the Referees is structured in a clear and easy-to-follow sequence: (1) comments from Referees; (2) author's response; (3) author's changes in manuscript.

The manuscript "Harmonization and comparison of vertically resolved atmospheric state observations: methods, effects, and uncertainty budget" by A. Keppens et al. presents a summary of different methods for harmonization and comparison of atmospheric profile retrievals. This paper summarizes different approaches to harmonize data using a consistent mathematical terminology, which makes it easy to contrast different methods and estimate a contribution of each manipulation to the final result.

[Figure]

This study fits well to the scope of the problems discussed in the AMT. This manuscript would be a good addition to the special TUNER (Towards Unified Error Reporting) issue laying a theoretical base for the atmospheric profile comparisons and error analysis. The results are presented in a consistent manner. The manuscript is very technical, but overall it is well written. I would recommend this paper for the publication in AMT after some technical corrections. My general and technical comments are listed below.

General comments: (1) Page 4, line 5: Do you mean cross-covariances here – the covariances between different error terms? Please, re-phrase this statement. (2) The current statement – "This expression assumes that the covariances of the difference error terms are not considered in the ex-ante covariance matrices" – could indeed be misleadingly interpreted. The error terms however are expected to be independent, as now stated explicitly. What is meant here is that some of the difference error estimates are sometimes provided within the ex ante uncertainty covariance matrix. If this is the case, the corresponding term should obviously be removed from Eq. (7). The authors have rephrased the first sentence below Eq. (7) to make this more clear. (3) The first sentences below Eq. (7) has been rephrased as follows: "This expression assumes that the covariance matrices of the difference error terms are independent and are not already included in the ex-ante covariance matrices. Eq. (7) should be corrected for those which are."

(1) Page 6, lines 16-19: I am not sure if I understand this part of the manuscript. To interpolate a vertical profile from a coarser vertical scale to a finer scale, one needs to assume an underlying function (linear, quadratic etc.). This interpolation might introduce an additional source of error into the comparison results that authors did not mentioned and considered here. If one degrades vertical resolution of the profile reported on a finer vertical scale, then no additional information is required. Typically, it is recommended to make comparisons on a vertical scale of the measurement with a coarser resolution. There is a discussion of effect of interpolation in section 5, but it would be good if you can mention that here as well. (2) The authors agree that "If

one degrades vertical resolution of the profile reported on a finer vertical scale, then no additional information is required." This however only generally holds when the target grid is a subset of the input grid of the regridding operation. If the target grid is not a subset of the input grid, then an underlying function is required nevertheless, just as for the coarse to fine scale regridding. The fine to coarse scale regridding however always loses information as intermediate points are removed. This is hardly the case for coarse to fine regridding. E.g., linear interpolation only inserts information-neutral grid points. The recommendation "to make comparisons on a vertical scale of the measurement with a coarser resolution" is agreed with and has therefore been included. (3) The first sentence "Straightforward regridding by (linear or other) interpolation only appropriately works when going from a coarser-resolution input grid to a finer-resolution target grid." has been extended as follows: "Straightforward regridding by (linear or other) interpolation only appropriately works, i.e. with minimum information loss, when going from a coarser-resolution input grid to a finer-resolution target grid." And at the end of the paragraph, the following sentence has been added: "The elements of W are determined by the interpolation function one applies, which introduces an additional term in the uncertainty budget (see Section 6)." Additionally, the next bullet now starts with the following sentence: "In order not to suggest a vertical resolution that is misleadingly much higher than the effective vertical resolution of (one of) the observations, atmospheric state profile comparisons are often made on the vertical grid of the product with the coarsest sampling."

(1) Page 7, equation 13: Why do you apply two operators (max/min) simultaneously? How would these two operators work? In the original paper referenced here [Langerock et al., 2015], authors use min or max, but not a combination of the two. (2) It is agreed that equation 13 is not fully clear in its present form. In fact, the nested max and min in the two terms indicated the layers' upper and lower bounds respectively, and not the mathematical max or min (as for the first occurrences). This has now been made clear in the equation and the text. (3) The second max and min in each term have been replaced by an upper height index U and L, respectively, and the first sentence

following the equation has been extended with the following: "and the indices U and L indicating the layers' upper and lower height bounds, respectively"

(1) Page 19, Conclusions. Conclusions are not well written, and I would suggest you make some revision. I listed some suggestions below in the technical section. (2) The authors acknowledge the reviewer's account of the conclusions and have rewritten most of the middle part, largely in agreement with the reviewer's specific technical comments. (3) The middle part of the conclusions section has been rewritten as follows: "The harmonization of two profiles' representations is mandatory for data comparisons and for proper quantitative $\chi^2$ testing of the resulting total difference covariance. Other data manipulations are needed to reduce the uncertainty budget of the comparison by minimizing the contributions due to differences in retrieval characteristics and spatiotemporal co-location. Ten matching operations have been identified and expressed in a consistent way using common matrix algebra. These operations include procedures for converting the ex-ante covariance matrix and the averaging kernel matrix (for retrieved products) associated to each atmospheric profile. It has therefore also been discussed how the information content of a retrieved product, as calculated from its AKM, is modified by each harmonization operation. Finally, it has been examined which terms of the error covariance are removed from the full comparison uncertainty budget by each harmonization operation"

Technical comments: (1) Page 1, line 11: the reference is missing. (2) A reference was not intended here. Rather, some measurements that are used in comparisons are called "reference measurements". The adjective was put between brackets to indicate both 'regular' measurements and 'reference' measurements. The authors believe that this is clear from the current text. (3) No action has been taken.

(1) Page 2, lines1-2: there are many abbreviations here that have not been introduced earlier in the text. (2) The authors agree that the abbreviations used might not be clear to all readers. The phrasing of this sentence has been changed, with all abbreviations expanded or removed, and a reference added. (3) The phrasing has been changed

into the following: "Carried out in the context of several satellite validation studies (for Sentinel-5p, the European Space Agency's Climate Change Initiative, the Satellite Application Facility on Atmospheric Composition Monitoring) and of the exploration of advanced data fusion methods (Cortesi et al., 2018)..." The reference to Cortesi et al. (2018) has been added.

(1) Page 3, lines 1-10: Typically, Xa and Sa are used to define a priori profile and corresponding matrix. It might be easier for readers to follow your paper if you use the established terms. (2) Xp and Sp resulted from an in-house notation, but using Xa and Sa is equally valid, does not conflict with other notations, and might indeed better match general practice. Subscripts of prior-related vectors and matrices have therefore been changed. (3) Xp, Sp, and Rp (and variations thereof) have been replaced by Xa, Sa, and Ra.

(1) Page 4, line 18: It might be better to re-phrase: "a full assessment and quantification of all contributions to the difference error Sdelta are necessary" (2) The authors are grateful for this nice suggestion. The text has been adapted accordingly. (3) "a full assessment and quantification of all difference error contributions to S_delta is necessary" has been replaced by "a full assessment and quantification of all contributions to the difference error S_delta are necessary"

(1) Page 5, lines 6-7: I suggest to replace this phrase with "the latter may introduce a bias and increase the spread due to uncertainties in the ancillary data" (2) The original phrasing "the latter may introduce both manipulation (difference) and ancillary data uncertainties" could indeed be improved. The suggestion by the reviewer however neglects uncertainties due to data manipulations. The reviewer's suggestion has therefore been extended in order to be in agreement with the full initial meaning of the statement. (3) The initial phrasing has been replaced by the following, more general statement: "A representation conversion may introduce a bias and reduce the precision due to uncertainties in the ancillary data and data manipulations..."

(1) Page 5, eq. 8 (also eq. 12): there are three equations written in one line. I suggest to have one equation in each line and have (a, b, c) labels for each equation. (2) The authors agree. (3) Eqs. (8) and (12) have been subdivided into Eqs. (8a) to (8c) and (12a) to (12c), respectively.

(1) Page 7, lines 29-30: the sentence needs some re-wording, because the meaning is not clear. (2) The substance "As any inversion approach of a retrieval outcome's vertical smoothing results in an ill-posed deconvolution..." might indeed be unclear to the non-expert reader. The authors have simplified this statement. (3) The original wording has been replaced by the following: "As the algebraic inversion of a retrieved profile's vertical smoothing is typically an ill-posed problem..."

(1) Page 19, line 1: I suggest to re-phrase with "The harmonization of a pair of atmospheric profile retrievals and their representations is required" (2) In the initial phrasing the harmonization of the profiles' representations was specifically intended. As this is different from the suggestion by the reviewer, the authors prefer keeping the text as is. (3) No changes have been made based on this comment.

(1) Page 19, lines 3-4: This sentence needs some re-wording. Maybe something like "Other data manipulations are needed to reduce the error budget of the comparison by minimizing contributions due to differences in retrieval characteristics and spatiotemporal sampling." (2) This suggestion has been included, with "sampling" replaced by the more general "co-location" in order to also include smoothing. (3) See text update at the end of the general comments section.

(1) Page 19, lines 4-5: replace with "have been identified and expressed in a consistent way using common matrix algebra." (2) This suggestion has been included. (3) See text update at the end of the general comments section. Additionally "from the literature" has been added after "identified".

(1) Page 19, lines 5-6: Suggest to replace with "These operations include procedures for converting the ex-ante covariance and averaging kernels matrices associated to

each atmospheric profile retrieval." (2) This suggestion has been included. (3) See text update at the end of the general comments section.

(1) Page 19, lines 7-10. It would be easy to read and understand if you simply say "In this study we discussed" or "Finally, we examined" (2) The authors agree with the reviewer that sometimes the active voice simplifies an expression of accomplished work. Only the passive voice however has been used throughout this manuscript, and the authors therefore prefer to stick to this voice also in the conclusions. The original phrasing however has been simplified to increase readability. (3) The initial phrasing of the first sentence has been replaced as follows: "Therefore the effect of each harmonization operation on the information content of a retrieved product, as calculated from its AKM, has also been discussed."

(1) Page 19, lines 8-10. I would rather say "what terms of the error covariance are removed" (2) This suggestion has been included (with "what" replaced by "which"). (3) See text update at the end of the general comments section.